# UNSUPERVISED EXPECTATION LEARNING FOR MULTISENSORY BINDING

## ABSTRACT

Expectation learning is a continuous unsupervised learning process which uses multisensory bindings to modulate unisensory perception. As humans, we learn to associate a barking sound with the visual appearance of a dog, and we continuously fine-tune this association over time, as we learn, e.g., to associate high-pitched barking with small dogs. In this work, we address the problem of building a computational model that captures two important properties of expectation learning, namely continuity and the lack of any external supervision other than temporal co-occurrence. To this end, we present a novel hybrid neural model based on audio/visual autoencoders and a recurrent self-organizing network for stimulus reconstruction and multisensory binding. We demonstrate that the proposed model is capable of learning concept bindings, i.e. dog barking with dogs, by evaluating it on unisensory classification tasks for audi-visual stimuli using the 43,500 Youtube videos in the animal subset of the AudioSet corpus. In addition, our analysis and discussion explain how the expectation learning mechanism enforces the generation of high-level bindings and how they contribute to audio-visual recognition.

## 1 INTRODUCTION

Multisensory binding is one of the most important processes that humans use to understand their environment. By using different sensory mechanisms, we are able to collect and process distinct information streams from the same experience which leads to a highly complex association learning. This mechanism allows us to improve the perception of individual stimuli (Frassinetti et al. (2002)), solve contextual, spatial and temporal conflicts (Diaconescu et al. (2011)), and progressively acquire and integrate novel information (Dorst & Cross (2001)).

An important process of multisensory binding is known as the expectation effect (Yanagisawa (2016) (2016)). When perceiving an event, we compare it to other events we have experienced before, and make certain assumptions based on our experience. For instance, when seeing a cat, we expect it to meow and not to bark. This effect modulates our multisensory association in terms of top-down expectation. In consequence, when a cat barks to us, we assume that our perception is inconsistent, and assume that either unisensory perception failed, the spatial or temporal attention was misleading, or we create a new species of a barking cat. For each of these scenarios, our brain adapts to the situation and we update our multisensory knowledge. This learning process referred to learning by expectation (Ashby & Vucovich (2016)), strongly suggests the role of unsupervised learning for multisensory binding, and leads to an adaptive mechanism for learning of novel concepts (Ellingsen et al. (2016)).

In this work, we propose a novel method that mimics the human ability to perform expectation learning and ask the following questions:

Q.1 *How can we build a computational model that allows for continuous unsupervised learning of multisensory bindings?*

Q.2 *Can we adapt the expectation learning from humans to this model and use it to generate expected unisensory visual stimuli from auditory stimuli and vice versa?*

Q.3 *Can we exploit the generated expected stimuli to improve unisensory classification?*

In Section 2 we investigate several other multisensory integration approaches and identify a lack of methods that are both unsupervised and continuous. To address Q.1, we employ autoencoders to learn auditory and visual representations, which allows for unsupervised learning. As a novelty and core mechanism to address continuity, we propose to link the autoencoders with a recurrent Grow-When-Required (GWR) neural network that changes its size as demanded, thus allowing for the continuous learning of multisensory binding (see Section 3). We address Q.2 by hypothesizing that the recurrent GWR network learns prototypes of multisensory bindings, which allows us to reconstruct auditory information from visual stimuli and vice-versa. For example, when perceiving the sound of a cat, we expect the model to reconstruct the image of a cat, while when a dog enters a scene, the sound of the dog will be reconstructed.

This expectation learning mechanism is described in Section 4. By extending the GWR association mechanism, we expect the model to be able to create concept-level bindings. Specifically, we hypothesize that by activating the neural units that represent prototypical concepts such as cats, dogs, and horses, the model will reconstruct prototypical auditory and visual stimuli in the absence of any sensory input. Our novel method is inspired by the multisensory imagery effect (Spence & Deroy (2013)), i.e., the ability of humans to simplify absent stimuli into concepts, and to use the abstract concepts to reconstruct unisensory information to enhance the overall perception. with Q.3 we ask whether this effect can be used to improve the classifiaction performance and hypothesize that our approach improves unisensory classification by reconstructing unisensory stimuli based on multisensory bindings.

To the best of our knowledge, there exists no standard benchmark to evaluate audio-visual bindings. Therefore, we propose a series of experiments to measure how far the expectation learning mechanism improves unisensory classification. To this end, we employ the Youtube AudioSet corpus (Gemmeke et al. (2017)), which contains human-labeled samples of Youtube videos based on the audio information. We select the animal subset of the corpus consisting of 44k samples to train the multisensory bindings in an unsupervised manner. We then exploit the multisensory bindings by using them to train a classifier for 24 different animal classes. We then employ the classifier to recognize absent stimuli, i.e., recognizing auditory stimuli when visual stimuli is present and vice-versa. In Section Section 5 we summarize our efforts to perform a fair and thorough evaluation of our model. We perform the experiments in three steps and assess the contribution of each step of the expectation and binding mechanism to compare it with state-of-the-art solutions for unisensory classification. In Section Section 6 we summarize the results of our experiments. We show that the expectation learning improved the recognition of unisensory stimuli and it presents competitive performance with state-of-the-art solutions.

To confirm our hypotheses, we discuss the results in Section 7, providing evidence that correlates our network behavior with the multisensory imagery effect. Also, we discuss the capabilities and limitations of our model. We conclude in Section 8 that the expectation learning mechanism improves the quality of the multisensory association by providing a better unisensory classification.

## 2 RELATED WORK

Most existing computational models for multisensory learning apply explicit weighted connections while integrating the sensor information which are learned using early (Wei et al. (2010)) or late (Liu et al. (2016); de Boer et al. (2016)) fusion techniques. These weighted connections are usually tuned in a data-driven manner, whereby the data distribution directly affects the multisensory binding. Such existing methods have the drawback that they require supervision and that they are sensitive to the training data distribution when performing the multisensory integration. More accurate models apply the neurophysiological findings on unisensory biasing for multisensory computational models (Pouget et al. (2002); Rowland et al. (2007); Kayser & Shams (2015)). Such models, although similar to the brain's neural behavior, are usually not feasible to be used on real-world data, as they are mostly applied to simple stimuli scenarios, and do not scale well. There exist other complex models that implement attention mechanisms based on multisensory information, but the most recent focus in this area is on data-driven fusion models (Hori et al. (2017); Barros et al. (2017); Mortimer & Elliott (2017)). The introduction of expectation learning would give these models the ability to adapt better to novel situations and learn with its own errors in an online and continuous way.

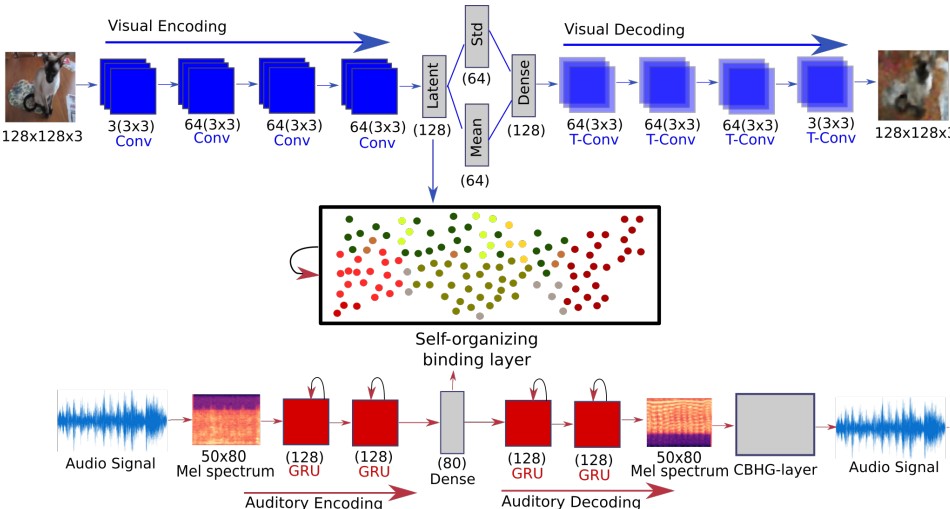

Figure 1: An overview of the proposed model with the audio/visual autoencoder structures and the self-organizing binding layer.

Recent contributions make use of data-driven learning for multisensory representations Arandjelovic & Zisserman (2017); Senocak et al. (2018); Owens & Efros (2018); Kim et al. (2018). Such solutions make use of different transfer learning and attention mechanisms to improve unisensory recognition and localization. Although they present impressive result in this specific tasks, they still rely on strongly labelled data points or are not suitable for online learning given that they have a extensive training proceeding. In particular, the work by Arandjelović & Zisserman (2017) introduces a data-driven model for multisensory binding with bottom-up modulation for spatial attention. Their model uses the network's activity to spatially identify which part of an image a certain sound is related to. Although the model is data-driven, the authors claim that it learns real-world biasing on a multisensory description for unisensory retrieval by using a large amount of real-world training data. Their results show that the model can also use a number of unisensory channels to compensate absent ones and identify congruent and incongruent stimuli. Overall, this approach has produced impressive results on spatial location and cross-sensory retrieval.

A similar approach was presented by Zhou et al. (2017), but focused on audio generation. Their model relies on a sequence-to-sequence generator to associate audio events to visual information. The same generator is use to generate audio for newly presented video scenes. This requires an external teacher to identify congruent and incongruent stimuli which makes it impossible to be used in online learning scenarios. Also, all of them are deeply dependent on an end-to-end deep learning strategy. Furthermore, it cannot learn novel information without the need of extensive retraining the entire model again.

## 3 MULTISENSORY TEMPORAL BINDING

To reconstruct auditory and visual stimuli, we developed neural networks based on autoencoders for each of the unisensory channels. These networks encode high-dimensional data into a latent representation and reconstruct real-world audio-visual information. The binding between auditory and visual information is realized by means of a recurrent GWR network. The GWR is a self-organizing network that learns to create conceptual prototypes of a data distributions in an unsupervised, incremental learning manner. To address the temporal aspects of coincident binding, we extend the Gamma-GWR (Parisi & Wermter (2017)) which endows prototype neurons with a number of temporal contexts to learn the spatiotemporal structure of the data distribution. An overview of our model is illustrated in Figure 1

### 3.1 VISUAL CHANNEL

To process high-level information in the visual channel, we realized a variational autoencoder (VAE) (Kingma & Welling (2013)) which enforces the encoded latent variables to follow a Gaussian distribution. The VAE produced better results when recognizing the animals from the AudioSet dataset when compared to normal deep autoencoders. We hypothesize that this improvement is due to the great variance on the image quality, perspective and resolution of the visual information of our dataset. Most likely the VAE learns to represent the important characteristics of the animals through the latent vector sampling, instead of learning to reconstruct the entire encoded image.

To train the VAE, we implemented a composite loss function based on the image reconstruction error and the Kullback-Leibler (KL) divergence between the encoded representation and a the Gaussian unit. This composite loss function is important to enforce that the encoded representations learn general concepts of the animals, instead of reconstructing input images from memorized parameters.

Our model receives as input a 128x128 color image. The input data is processed by our encoding architecture which is composed of a series of four convolution layers, with a stride of 2x2, and kernel sizes of dimension 3x3. The first convolution layer has three channels and the subsequent three layers have 64 filters. The latent representation starts with a fully connected layer with 128 units. We compute the standard deviation and mean of this layer's output, generate a Gaussian distribution from it and sample an input for another fully connected hidden layer with 64 units, which is our final latent representation. The decoding layer has the same structure as our encoding layer but in the opposite direction and applying transpose convolutions.

### 3.2 AUDITORY CHANNEL

For the auditory channel, we implement a recurrent autoencoder based on Gated Recurrent Units (GRU) (Cho et al. (2014)). Recurrent units allow us to reconstruct audio with better quality than using non-recurrent layers since auditory signals are sequential, and each audio frame depends highly on previous contextual information.

As input and output of the auditory autoencoder, we compute a Mel spectrum which we generate from the raw waveform. To reconstruct the audio from the output Mel spectrum, we employ a convolutional bottleneck CBHG network model (Lee et al. (2017)) which consists of a 1-D convolutional bank, a highway network and a bi-directional GRU layer. This network receives as input the Mel spectrum, and outputs a linear frequency spectrum which is then transformed into waveform using the Griffin Lim algorithm. This approach of transforming Mel coefficients to a linear spectrum and then to waveform achieved better audio synthesis quality than performing Griffin Lim on the Mel spectrum directly (Wang et al. (2017)), and it improves the audio data of our expectation learning approach.

We performed hyperparameter optimization for the autoencoder and found that an audio spectrum window length of 50ms, a window shift of 12.5 ms with 80 Mel coefficients and 1000 linear frequencies yield best results. We also found that 80 units for the dense bottleneck layer and two GRU layers with 128 units each for both the encoder and decoder network are sufficient for achieving a high audio quality. An additional number of Mel coefficients, GRU layers, and neural units did not significantly improve the reconstruction quality. The number of bottleneck units is important for the multisensory binding as it determines the number of connections between the binding layer and the audio encoder and decoder.

### 3.3 SELF-ORGANIZING TEMPORAL BINDING

To learn coincident bindings between audio and visual stimuli, we use an unsupervised binding layer. This layer is implemented as a recurrent GWR network, which is an unsupervised model for learning spatiotemporal prototype representations. The layer receives as input the latent representation of our visual and auditory channels which occur coincidentally. To synchronize the two data streams, we resample video and audio streams to a temporal resolution of 20 frames per second, i.e., each video frame is associated with 12.5 ms of auditory information. In contrast to traditional self-organizing models with winner-take-all dynamics for the processing of spatial patterns, the Gamma-GWR (Parisi & Wermter (2017)) computes the winner neuron taking into account the activity of the network for the current input and a temporal context. Each neuron of the map consists of a weight

vector $\mathbf{w}_j$ and a number $K$ of context descriptors $\mathbf{c}_j^k$ (with $\mathbf{w}_j, \mathbf{c}_j^k \in \mathbb{R}^n$). As a result, recurrent neurons in the map will encode prototype sequence-selective snapshots of the input. Given a set of $N$ neurons, the best-matching unit (BMU), $\mathbf{b}$, with respect to the input $\mathbf{x}(t) \in \mathbb{R}^n$ is computed as:

$$b = \arg\min_{j \in N} \left( \alpha_0 \|\mathbf{x}(t) - \mathbf{w}_j\|^2 + \sum_{k=1}^{K} \alpha_k \|\mathbf{C}_k(t) - \mathbf{c}_{j,k}\|^2 \right), \tag{1}$$

$$\mathbf{C}_k(t) = \beta \cdot \mathbf{w}_{I(t-1)} + (1-\beta) \cdot \mathbf{c}_{I(t-1),k-1}, \tag{2}$$

where $\alpha_i$ and $\beta \in (0;1)$ are constant values that modulate the influence of the current input with respect to previous neural activity, $\mathbf{w}_I(t-1)$ is the weight of the winner neuron at $t-1$, and $\mathbf{C}_k \in \mathbb{R}^n$ is the global context of the network ($\mathbf{C}_k(t_0) = 0$).

New connections are created between the BMU and the second-BMU of an input. When a BMU is computed, all the neurons the BMU is connected to are referred to as its topological neighbors. Each neuron is equipped with a habituation counter $h_i \in [0,1]$ expressing how frequently it has fired based on a simplified model of how the efficacy of a habituating synapse reduces over time. In the Gamma-GWR, the habituation rule is given by $\Delta h_i = \tau_i \cdot \kappa \cdot (1 - h_i) - \tau_i$, where $\kappa$ and $\tau_i$ are constants that control the decreasing behavior of the habituation counter (Marsland et al. (2002)). To establish whether a neuron is habituated, its habituation counter $h_i$ must be smaller than a given habituation threshold $h_T$. The network is initialized with two neurons and, at each learning iteration, it inserts a new neuron whenever the activity of the network $a(t)$ of a habituated neuron is smaller than a given threshold $a_T$, i.e., a new neuron $r$ is created if $a(t) < a_T$ and $h_b < h_T$. The training of the neurons is carried out by adapting the BMU $b$ and its topological neurons $n$ according to:

$$\Delta \mathbf{w}_i = \epsilon_i \cdot h_i \cdot (\mathbf{x}(t) - \mathbf{w}_i), \tag{3}$$

$$\Delta \mathbf{c}_{k,i} = \epsilon_i \cdot h_i \cdot (\mathbf{C}_k(t) - \mathbf{c}_{k,i}), \tag{4}$$

where $\epsilon_i$ is a constant learning rate. The learning process of the Gamma-GWR is unsupervised and driven by bottom-up sensory observations, thereby either allocating new neurons or adapting existing ones in response to novel input. In this way, fine-grained multisensory representations can be acquired and fine-tuned through experience.

As an extension of the Gamma-GWR, we implement temporal synapses for the purpose of predicting future frames from an onset frame. We implement temporal connections as sequence-selective synaptic links that are incremented between those two neurons that are consecutively activated. When the two neurons $i$ and $j$ are activated at time $t-1$ and $t$ respectively, their synaptic link $P_{(i,j)}$ is strengthened. Thus, at each learning iteration, we set $\Delta P_{(I-1,b)} = 1$, where $I-1$ and $b$ are respectively the indexes of the BMUs at time $t-1$ and $t$. As a result, for each neuron $i \in N$, we can retrieve the next neuron $v$ of a prototype sequence by selecting

$$v = \arg\max_{j \in N \setminus i} P_{(i,j)}. \tag{5}$$

This approach results in the learning of trajectories of neural activations that can be reconstructed in the absence of sensory input.

## 4  EXPECTATION LEARNING

As the self-organizing layer is updated in an unsupervised Hebbian manner, it learns to associate audio-visual stimuli online. That means that the binding process is entirely data-driven, without the necessity of supervision. More specifically, after finding the BMU related to a unimodal perceived stimulus, the associated absent stimuli will be reconstructed based on the prototypical concept that this neuron learned. This is possible because each neuron in the self-organizing layer processes the union of the auditory and visual encodings at training time, where both signals are provided.

This reconstruction and expectation learning capability is the basis for our following proposal of a re-training mechanism for the self-organizing layer. We first pre-train our self-organizing binding to generate prototype neurons with strong audio-visual encodings. This allows the model to learn a prior association between auditory and visual concepts. After the network has learned these associations, we use unknown data to fine-tune the bindings with the expectation learning.

We first encode a visual or auditory stimulus ($s$), and compute the BMU ($b_{av}$) using only the associated auditory or visual weights:

$$b_{av} = \arg\min_{j \in N} \left( \alpha_0 \|\mathbf{s}(t) - \widetilde{\mathbf{w}}_j^s\|^2 + \sum_{k=1}^{K} \alpha_k \|\widetilde{\mathbf{C}}_k^s(t) - \widetilde{\mathbf{c}}_{j,k}\|^2 \right),$$ (6)

where $\widetilde{\mathbf{w}}_j^s$ represents the audio or visual representation encoded on the neuron's weights. In this case, the global context of the network at any time step ($\widetilde{\mathbf{C}}_k^s(t)$) is represented by the stimulus encoding, the same happens with the BMU context ($\widetilde{\mathbf{c}}_{j,k}$). We then use the auditory and vision parts of the multisensory representation stored on $b_{av}$ to reconstruct the auditory ($a'$) and visual ($v'$) information using the specific channel decoding $D_v$ for vision and $D_a$ for audio:

$$\begin{aligned} a' &= D_a(b_a), \\ s' &= D_v(b_v). \end{aligned}$$ (7)

By doing this for both perceived auditory and visual signals, we create two extra pairs of multisensory stimuli by combining the perceived auditory and visual ones with the reconstructed auditory and visual. We bind the encoded information of the reconstructed audio-visual information with the original perceived stimuli and re-train the self-organizing layer with the new pairs. By pairing the perceived and the reconstructed stimuli representations, we enforce the self-organizing layer to learn concepts and not instances of the animals. In consequence, animals which sound similar will be paired together, and connections of coincident stimuli will be learned with relatively small amounts of training data. Incongruences will cause the model to pair different audio-visual stimuli, thus creating new prototype neurons, but these will be forgotten quickly by the self-organizing layer as they occur less frequently.

## 5 EXPERIMENTAL SETUP

Our goal is to evaluate the performance of the model to reconstruct audio/visual stimuli based on unimodal perception. Also, we intend to evaluate the conceptual relations learned by the network. Although there exists several datasets with multimodal information, the animal subset of the AudioSet corpus (Gemmeke et al. (2017)) presents a unique advantage for our evaluation: It contains natural scenarios with different levels of conceptual binding, including broader prototype associations like images of cats linked to meowing, but also more fine-grained associations like high-pitched barking liked to with small dogs.

Each video in the dataset has a duration of 10s and it is possible that, e.g., there is both a cat and a dog present in the video. As there are no standard published results on this specific task for the AudioSet corpus, we run a series of baseline recognition experiments that serves as main comparison to measure our model's performance. To obtain a precise measure of the contribution of the expectation learning, we decided to cluster some overlapping classes and use 16 single labels, one per video: Cats ("Cat" + "Meow"+ "Purr"), Dogs ("Bark"+"Dog"+"Howl"), Pigs ("Oink" + "Pig"), Cows ("Moo"+"Cattle, bovinae"), Owls ("Owl" + "Coo"), Birds, Goats, Bee ("Bee, was, etc.."), Chickens ("Chicken, rooster"), Ducks ("Duck"), Pidgeons ("Pidgeon, dove"), Crows ("Crow"), Horses ("Horse"), Frogs ("Frogs"), Flies ("Fly, housefly"), Lions ("Roaring cats (lions, tigers)"). We used the unbalanced training subset consisting of approximately 43.500 videos to train our model and evaluated it with the test subset consisting of approximately 20.000 videos. The labels of this dataset were crowdsourced based on the video descriptions.

We perform two sets of experiment: one to evaluate the contribution of the expectation learning to the multisensory binding and one to compare the performance of our model with currently successful deep learning models for unisensory recognition.

The first set of experiments is divided into three steps. In $Exp_1$ we train the multisensory bindings of the GWR using half of the training subset in order to guarantee that the model learned strong audio-visual prior bindings. In $Exp_2$ we continue the training of the $Exp_1$ network using the other half of the training subset. This experiment serves as a baseline for learning bindings without expectation

Table 1: Mean accuracy, in percentage, and standard deviation of our experiments: the baselines results, training the network with half and all the samples of the training set and reconstructing the absent modality using the expectation mechanism.

| Model | Audio | Vision |
|---|---|---|
| $Exp_1$ - Prior binding association | 58.5 (3.1) | 69.0 (3.9) |
| $Exp_2$ - Without expectation | 66.4 (2.4) | 86.8 (3.2) |
| $Exp_3$ - With expectation | 70.8 (3.2) | 89.8 (1.9) |
| Inception V3 (Ioffe & Szegedy (2015)) | - | 90.4 (1.3) |
| SoundNet (Aytar et al. (2016)) | 68.5 (2.4) | - |

and as a main comparison point for the contribution of the expectation learning mechanism. And finally, $Exp_3$, where we repeat the continuation of the training of the $Exp_1$ network with the other half of the training subset but now using the expectation learning mechanism when creating the GWR associations.

To evaluate the performance contribution of each of our experimental steps on the association learning, we implemented a supervised classifier for each of the channels channel (auditory and visual). It receives as input the encoded representation from the GWR of a perceived audio or visual stimuli. To evaluate the capability of the model to learn meaningful associations, we classify always an absent stimuli, i.e. when perceiving an auditory stimuli, the network uses the associated visual stimuli as input to the classifier and vice-versa. That means that when perceiving 50ms of audio, we have an associated representation of 4 frames and vice-versa. As the videos from the AudioSet dataset have a lenght of 10s, we use a simple voting scheme to obtain the final label. For every 50ms for audio and every 4 frames for video we produce one label and after having all the labels for a 10s video, we select the one which appears more often.

Each classifier is composed of a dense layer with 128 units and an output softmax layer. We optimized our autoencoders and classifier to maximize the recognition accuracy on the training subset using a tree-structured Parzen estimator (TPE) (Bergstra et al. (2011)) and use the optimal parameters through all of our experiments.

Our second set of experiments was designed to evaluate how our proposed model compares to deep learning networks for auditory and visual stimuli recognition. We compare our model with the Inception V3 network (Ioffe & Szegedy (2015)) for the visual stimuli, and the SoundNet (Aytar et al. (2016)) for the auditory stimuli. These models present competitive results on different audio-visual recognition tasks Kiros et al. (2018); Jiang et al. (2018); Kumar et al. (2018); Jansen et al. (2018). Herein, our goal is not to propose the best benchmark model for audio-visual recognition, but to assess the contributions of the expectation learning on the reconstruction of absent stimuli.

For all experiments, we trained the models 10 times and report the mean accuracy and standard deviation for each modality. We used the same 10% of the training subset as a validation set for each experiment, and used an early stopping mechanism based on the accuracy of the validation subset to prevent overfitting.

## 6 RESULTS

Our final results are depicted in Table 1. Our first experiment, $Exp_1$, demonstrates that training the model with half of the data, to create strong binding associations, was enough to obtain a baseline performance. Continuing to train the model using standard GWR associations ($Exp_2$) gives as an, expected, improvement of 8% of recognition accuracy for audio and more than 17% for vision when compared to $Exp_1$ experiment. The results of $Exp_3$ shows that the expectation mechanism improved the recognition of unisensory stimuli, when compared to $Exp_2$. We obtained an improvement of more than 4% on audio and 3% on vision.

The performance of the network follows the general behavior of other models to recognize vision stimuli better than auditory stimuli. This effect is demonstrated by the results of the SoundNet and Inception-V3 models. This is probably due the dataset presenting challenging audio stimuli with much background noise.

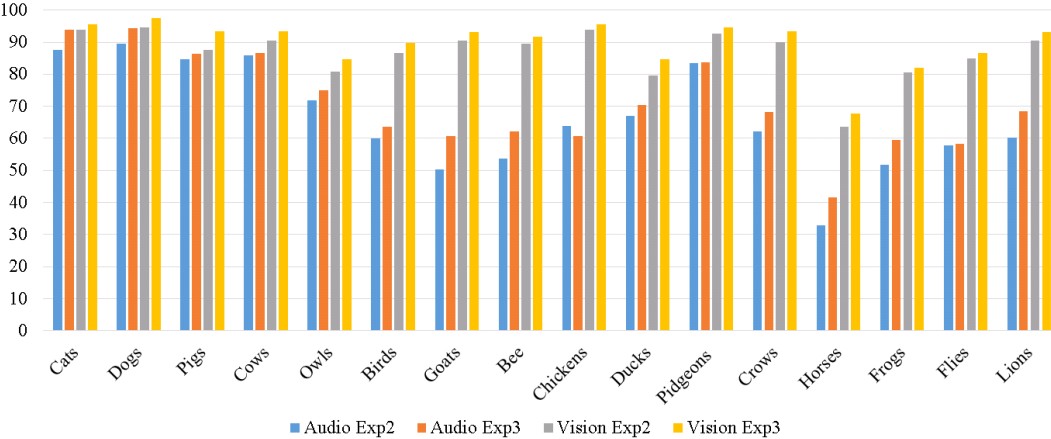

Figure 2: Mean accuracy per class, in percentage, of the reconstructed absent stimuli. We compare audio and visual reconstruction with the results when training the network with all the samples of the training set.

When compared to SoundNet and Inception-V3, our expectation model (represented by $Exp_3$ experiment) presents better auditory recognition, and comparable vision recognition performance. The auditory stimuli is more affected, as it presents much more noisy information. The network then relies more on the visual stimuli and creates neurons with strong visual encoding. This effect is represented by creating neurons with similar visual encoding associated to different auditory encoding. When training with expectation learning, the network created an average of 5400 neurons, while training without the expectation it created 4000 neurons.

The latent representation from the auditory and visual channels encode different characteristics of the stimulus, which are then bound by our self-organizing layer. The expectation learning enforced the generation of robust bindings, especially for distinct animals. The network ended up creating specific neurons for cats and dogs, and shared neurons for chickens and ducks, for example. This explains the improvement on the recognition of the reconstructed stimuli of easily separable animals, as illustrated in Figure 2.

# 7 DISCUSSION

As the self-organizing layer is updated in an unsupervised manner, it learns to associate audio-visual stimuli online. Moreover, by activating the BMU related to a specific perceived stimulus, the associated absent stimuli can be reconstructed based on the concept that this neuron learned. However, the reconstructed data is, of course, not identical to the original data. For example, when processing an image of a dog, the network will reconstruct an appropriate barking sound, but not exactly the sound that this specific dog would make. This mimics precisely the multisensory imagery effect (Spence & Deroy (2013)) of humans, who tend to simplify and cluster absent stimuli when asked to reconstruct them. For example, every time one sees a small yellow bird, the person will expect it to sound very similar to the ones she/he has seen before. This is an important effect that helps our model to reconstruct concepts instead of specific instances.

To provide an indication of this effect, and as an additional indicator for multisensory concept formation, we perform an overlapping analysis to estimate how well the model is binding and clustering audio-visual information. To this end, we first train the model with the expectation learning mechanism and then we classify every single neuron of the GWR using both audio and visual classifiers which generate two labels for each neuron: one for auditory and one for visual information. The total overlap between visual and auditory labels for each prototype neuron in our self-organizing layer is 93%, suggesting that our prototype neurons are very concise when storing audio-visual information. Performing the same experiment on the network training without the expectation mechanism gave us an overlap of 85% of the neurons.

Figure 3: Example of the reconstruction output. The left image displays the audio reconstruction when the visual stimulus is perceived. The right image displays the vision reconstruction when the audio stimulus is perceived.

Another effect that we investigate is multisensory correspondence (Spence & Driver (2000)). The effect causes humans not only to associate dogs with barking, but also, more specifically, small dogs with high-pitched barking. The associations between the stimuli are continuously reinforced as perceptive stimuli are experienced.

We observed this effect in some examples where the variety of animals was higher, such as the dogs. We illustrate one of these examples in Figure 3. The figure depicts the reconstruction of visual information based on an auditory stimulus of different dogs barking. A high-pitch barking generated images related to a small dog. Furthermore, when the simultaneous barking of more than one dog is processed, the network generates an image of several dogs. We expect this effect to become more visible with larger datasets that contain more diverse samples.

One important limitation of our approach is that both multisensory imagery and multisensory correspondence only occurs when both auditory and visual stimuli can be understood and represented as a simplified concept. A human cannot reconstruct precisely the characteristics of how the voice of a person will sound when reading a text, for example. Our experiments demonstrate that our model learns to associate high-level animal concepts, and even multisensory correspondences, but would not be applied to reconstruct information that demands a much higher precision, i.e.person identification.

# 8 CONCLUSION

Multisensory binding is a crucial aspect of how humans understand the world. Consequently, the development of computational systems able to adapt this aspect into information processing is important to many research fields. An extensive number of models has been proposed that incorporate different aspects of multisensory binding. However, our approach combines several novelties. It combines a Grow-When-Required (GWR) network with autoencoders to realize continuous unsupervised expectation learning. In addition, we propose to exploit expectation learning by reconstructing stimuli that can be used as additional training data to generate a significant positive effect on perceptive tasks like classification. We are, therefore, the first to introduce a quantitative metrics to measure the quality of multisensory bindings, while at the same time providing a novel proof-of-concept for a data augmentation mechanism to improve the accuracy and performance of unimodal classification methods.

An interesting future research direction is to also address spatial expectation, because this would provide a complementary component to integrate contextual, temporal, and spatial correspondence. Realizing the transfer of learned multisensory bindings is another unexplored research area that we plan to investigate as a follow-up to this work.

ACKNOWLEDGMENTS

Given the double-blind review nature of ICLR, we will update the Acknowledgments after the reviewing process.

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

# 9 APPENDIX

## 9.1 TRAINING PARAMETERS

### 9.1.1 VISION CHANNEL

Table 2 exhibits all the important parameters used to train our visual channel. We used ADAM optimizer with an adaptive learning rate.

Table 2: Training parameters of the Vision channel

| Parameter | Value |
|---|---|
| Epochs | 200 |
| Batch size | 32 |
| Optimizer | ADAM (Adaptive Learning rate) |
| Initial learning rate | 0.05 |
| ADAM beta1 | 0.9 |
| ADAM beta2 | 0.999 |

### 9.1.2 AUDITORY CHANNEL

Table 3 exhibits all the important parameters used to train our auditory channel. We follow the same training procedure as the vision channel, and also use ADAM optimizer with an adaptive learning rate.

Table 3: Training parameters of the Auditory channel

| Parameter | Value |
|---|---|
| Epochs | 250 |
| Batch size | 32 |
| Optimizer | ADAM |
| Initial learning rate | ADAM (Adaptive Learning rate) |
| ADAM beta1 | 0.9 |
| ADAM beta2 | 0.999 |

### 9.1.3 SELF-ORGANIZING BINDING LAYER

Table 4 exhibits all the important parameters used to train our gamma Growing-When-Required (GWR) network. We use a small insertion threshold, which helps the network to maintain a limited number of neurons, reinforcing the generation of high-abstract clusters.

## 9.2 DETAILED EXPERIMENTAL RESULTS

### 9.2.1 BASELINE EXPERIMENTS

Table 5 details the mean accuracy of our baseline experiments (SoundNet and Inception V3) for each of the classes of the evaluation subset.

Table 4: Training parameters of the Auditory channel

| Parameter | Value |
|---|---|
| Epochs | 50 |
| Insertion threshold | 0.01 |
| Context size | 4 |
| Initial Gamma Weights | 0.64391426, 0.23688282, 0.08714432, 0.0320586 |
| $\beta_b$ | 0.5 |
| $\epsilon_b$ | 0.2 |
| $\epsilon_n$ | 0.003 |

Table 5: Detailed results in accuracy (in %) and standard deviation for our baseline experiments.

| Animal | SoundNet | Inception V3 |
|---|---|---|
| Cats | 90,2 (3.2) | 94,8 (2.4) |
| Dogs | 92,5 (4.1) | 96,7 (2.5) |
| Pigs | 80,7 (3.7) | 95,6 (3.4) |
| Cows | 83,8 (3.5) | 94,8 (1.7) |
| Owls | 71,8 (1.4) | 87,8 (1.0) |
| Birds | 62,7 (2.2) | 90,6 (3.6) |
| Goats | 60,2 (3.9) | 95,8 (2.1) |
| Bee | 63,1 (1.1) | 91,2 (4.7) |
| Chickens | 59,8 (3.0) | 96,8 (2.3) |
| Ducks | 68,7 (4.1) | 85,1 (1.7) |
| Pidgeons | 76,8 (2.6) | 92,5 (3.1) |
| Crows | 67,9 (1.8) | 91,3 (2.7) |
| Horses | 43,6 (3.7) | 69,8 (4.1) |
| Frogs | 57,8 (1.4) | 79,8 (2.5) |
| Flies | 53,1 (1.3) | 89,8 (1.9) |
| Lions | 63,5 (3.4) | 94,5 (2.5) |

## 9.3 Expectation learning Experiments

Table 6 exhibits the detailed accuracy per class when evaluating our expectation learning model with the evaluation subset. Here we detail the results based on reconstructed audio, when the audio is absent on the perceived stimuli, and on reconstructed vision, when the vision is absent. We detail the experiments with and without the expectation.

## 9.4 Examples

Figure 4 illustrates some selected examples of impact of the expectation learning. The image exemplifies cases where the network trained with expectation learning improved the recognition. Each row shows an input audio example (and the class associated to it), and the reconstruction of a network trained with and without the expectation.

Table 6: Detailed accuracy (in %) for our baseline experiments.

| Animal Class | Audio | | Vision | |
|---|---|---|---|---|
| - | Without Expectation | With Expectation | Without Expectation | With Expectation |
| Cats | 87,6 (3.2) | 93,8 (2.1) | 93,8 (1.9) | 95,6 (2.1) |
| Dogs | 89,5 (3.6) | 94,4 (2.9) | 94,6 (2.2) | 97,5 (1.8) |
| Pigs | 84,6 (3.2) | 86,5 (3.7) | 87,5 (1.4) | 93,4 (1.7) |
| Cows | 85,9 (4.1) | 86,7 (2.7) | 90,4 (1.6) | 93,4 (2.8) |
| Owls | 71,8 (3.7) | 74,9 (2.9) | 80,7 (1.8) | 84,7 (1.9) |
| Birds | 60,1 (2.6) | 63,7 (1.9) | 86,7 (4.7) | 89,7 (3.7) |
| Goats | 50,2 (1.6) | 60,7 (3.7) | 90,4 (2.8) | 93,2 (1.9) |
| Bee | 53,7 (2.7) | 62,1 (3.9) | 89,5 (2.7) | 91,7 (3.1) |
| Chickens | 63,8 (1.9) | 60,7 (2.1) | 93,8 (1.7) | 95,7 (1.9) |
| Ducks | 66,9 (1.9) | 70,5 (2.8) | 79,5 (1.6) | 84,6 (2.9) |
| Pidgeons | 83,6 (4.7) | 83,8 (2.6) | 92,6 (2.7) | 94,7 (2.9) |
| Crows | 62,1 (1.9) | 68,3 (2.2) | 90,1 (2.0) | 93,4 (2.8) |
| Horses | 32,8 (2.6) | 41,6 (3.9) | 63,7 (3.1) | 67,8 (1.8) |
| Frogs | 51,8 (3.7) | 59,4 (2.7) | 80,6 (2.7) | 82,1 (3.4) |
| Flies | 57,8 (3.0) | 58,3 (2.5) | 84,9 (1.6) | 86,7 (2.6) |
| Lions | 60,3 (2.9) | 68,5 (2.6) | 90,4 (2.4) | 93,2 (3.8) |

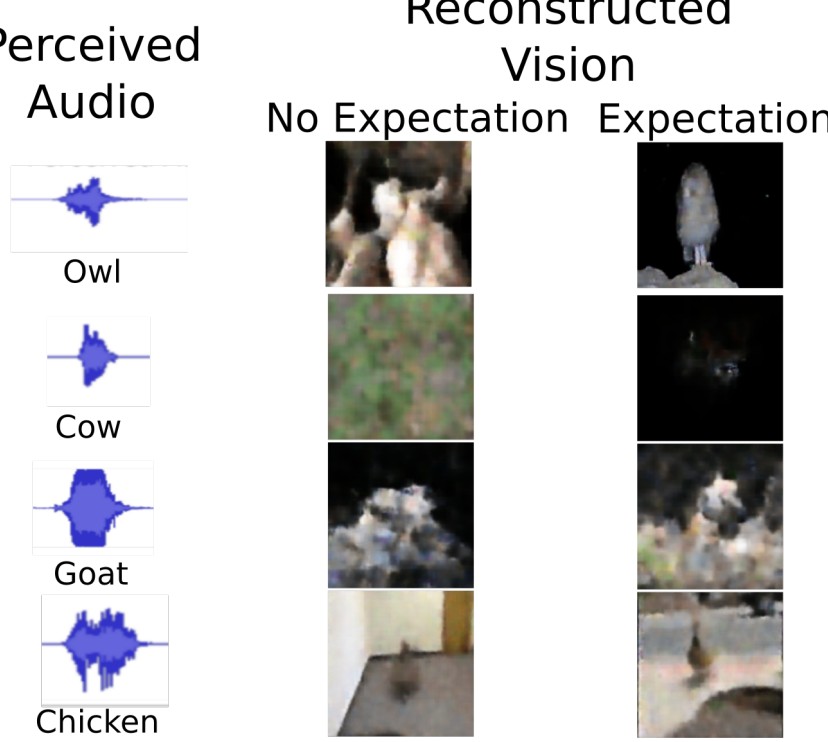

Figure 4: Examples of reconstructed images for the same perceived auditory stimuli of a network trained with and another network trained without expectation learning.

