# OpenReview forum: "Unsupervised Expectation Learning for Multisensory Binding"
_ICLR.cc/2019/Conference_

### Official Review · AnonReviewer1 · 2018-10-31
**interesting idea, but writing quality could be improved**

**Rating:** 5
**Confidence:** 2

**Review:**

The authors proposed an unsupervised learning framework to learn multisensory binding, using visual and auditory domain from animal videos as example. First, the visual and auditory inputs are autoencoded, and these latent codes are binding using a recurrent self-organizing network (Gamma-GWR). Furthermore, the authors proposed the expectation learning idea, which is inspire by psychology literature. In short, after the first pass of training using the real data. The authors fine tuned the model to bind the real data from one domain and the reconstructed data from another domain. This could be a good idea, as the authors pointed out, human usually bind all kinds of yellow bird to a same mental 'chirping' sounds. So, this expectation learning could potentially group the representation to a canonical one. Also, the authors showed in Table 1 that with the expectation learning, the model's recognition accuracy is improved a bit. I think it would be interesting to show the reconstruction output example (as in Fig. 3) for both model with and without expectation learning. To see if it is as the authors claim, that the model with expectation learning is reconstructing the missing modality with more canonical images/sounds. (This may not be the goal in other practice, though I'm convinced it is a potentially good psychological model as it explain well the multisensory imagery effect (Spence & Deroy, 2013).

I found this manuscript quite hard to follow though. The description seems sometime not flowing very smoothly. And there are some clear typos and mess up of math notations make the reading unpleasant. I have noted down several points below, and hope the authors could improve in the next iteration.

1. The description of variational autoencoder is not well written. The citation (Chen, 2016) is not the standard VAE paper people usually cite (unless the author is adopting something specific from the Chen's paper.). For example, the authors wrote "the KL divergence between the encoded representation and a sample from the Gaussian distribution" which sounds incorrect to me.

2. Why a Variational autoencoder is necessary for visual domain, but a regular autoencoder is used in auditory domain?

Typos:
1. page 2, 2nd line: a online --> an online
2. Use subscript I-1 to mean the winner neuron at t-1, I think this is not quite clear. I suggest to follow the notation in (Parisi & Wermter 2017), use I(t-1), which is easier to follow.
3. page 7, 2nd line: more than 17% for audio.  -> for vision.
4. page 8, 3rd line: not on the original high-abstraction data. Do the authors mean highly specific data? That seems make more sense.
5. Several notation mismatch here and there. for example, in formula 6 it is w_j^s, but in the text below it become w_{j,s}.

---

> ### Author Response · Authors · 2018-11-26
> **Improved model's description and examples**
>
> Reviewer 3
>
> Thank you for your points on our model's descriptions. As mentioned to the other reviewers, we updated the paper to reflect a better readability and understanding of our contributions. We also updated necessary and incorrect notations and descriptions. In particular, we answer your concerns as follows:
>
> 0. I think it would be interesting to show the reconstruction output example (as in Fig. 3) for both model with and without expectation learning. To see if it is as the authors claim, that the model with expectation learning is reconstructing the missing modality with more canonical images/sounds. (This may not be the goal in other practice, though I'm convinced it is a potentially good psychological model as it explain well the multisensory imagery effect (Spence & Deroy, 2013).
>
> R: We updated the appendix of the paper with an example of the reconstruction of visual stimuli for two networks: one with and one without the expectation learning. We choose examples where the expectation network presented the correct recognition to illustrate the differences in the bindings.
>
> 1. The description of variational autoencoder is not well written. The citation (Chen, 2016) is not the standard VAE paper people usually cite (unless the author is adopting something specific from the Chen's paper.). For example, the authors wrote "the KL divergence between the encoded representation and a sample from the Gaussian distribution" which sounds incorrect to me.
>
> R: We corrected our description of the VAE and how we hypothesize that the learned representations of the VAE improve our model. We also updated the VAE citation.
>
> 2. Why a Variational autoencoder is necessary for visual domain, but a regular autoencoder is used in auditory domain?
>
> R: The use of variational autoencoder for the visual domain was made purely based on the performance of the model. While we do not focus on optimizing all the possible hyperparameters of the model to extract every 0.1% of performance, our experiments showed a great improvement which could be ignored when using a variational autoencoder in contrast with a regular one. We theorize that the improvement was due to the great variance on the image quality, perspective and resolution of the visual information of our dataset. We updated this explanation on the Visual Channel section (Section 3.1) of our paper. Most likely the VAE learned how to represent the important characteristics of the animals through the latent vector sampling, instead of learning to reconstruct the entire encoded image.However, We did not explore fully our visual VAE to exemplify which are the learned characteristics and how they are separable on the latent distribution. We would include this on future work, as our focus on this paper was on the expectation learning mechanism.
>
> Typos:
> 1. page 2, 2nd line: a online --> an online
> 2. Use subscript I-1 to mean the winner neuron at t-1, I think this is not quite clear. I suggest to follow the notation in (Parisi & Wermter 2017), use I(t-1), which is easier to follow.
> 3. page 7, 2nd line: more than 17% for audio.  -> for vision.
> 4. page 8, 3rd line: not on the original high-abstraction data. Do the authors mean highly specific data? That seems make more sense.
> 5. Several notation mismatch here and there. for example, in formula 6 it is w_j^s, but in the text below it become w_{j,s}.
>
> R:  Thank you for your feedback. We updated the general readability of the paper regarding improving language and avoiding typos. In particular, we fixed the notation problems.

---

### Official Review · AnonReviewer3 · 2018-11-03
**a model that mimics expectation learning i.e. learning multisensory representations by training to predict the other modalities from a given modality**

**Rating:** 5
**Confidence:** 3

**Review:**

Overview and contributions: The authors introduce a model that mimics expectation learning (i.e. learning multisensory representations by training to predict the other modalities from a given modality, for example, image to audio, audio to image). The proposed model is based on an autoencoder structure with a recurrent self-organizing network for
multisensory binding of latent representations. The authors perform experiments to show the reconstruction of image and audio signals given the other, as well as discriminative results on audio and image classification.

Strengths:
1. The paper is well motivated by the point of view of human learning. I really liked the abstract and introduction!
2. I liked the recurrent self-organizing network presentation and usage.

Weaknesses:
1. While the paper is well motivated, I believe that the presence of multisensory expectation learning depends heavily on the type of multisensory data. For some modalities such as audio, it is clear what the mapping to language is (audio-to-language transcribing). For language-to-audio, there are multiple audio translations depending on the different tone of voice used by each person. Image-to-audio translation is also a one-to-many mapping. So I have concerns about how the model would work in these cases depending on the data used.
2. I don't believe that the proposed model achieves state-of-the-art results: from Table 1, image classification performance is outperformed by Inception V3, and for both modalities, I'm not sure why the authors did not compare with more recent baselines. The best audio baseline is from 2016...
3. It seems that this approach needs paired multisensory data for training, which limits the amount of training data as compared to unisensory models. Also, what if some sensors are noisy or missing? Is this model robust to such cases?

Questions to authors:
1. Refer to weakness points.
2. Can you comment on when you think multisensory expectation learning would work, and when it wouldn't? What types of data do we need, and from which modalities/sensors?

Presentation improvements, typos, edits, style, missing references:
1. Page 2: Our hybrid approach allowed -> Our hybrid approach allows
1. Page 7: audiotry recognition -> auditory recognition
2. Page 7: while improved the visual stimuli in 3% -> and improving ... by 3%
3. Multiple other typos and awkward phrasing, I would suggest the authors spend more time proof-reading their paper before submission.

---

> ### Author Response · Authors · 2018-11-26
> **Better discussion on the model's applicability**
>
> 1. R: The reviewer is correct on mentioning that multisensory expectation learning depends on the multisensory data modalities. When applied to audio translation, for example, expectation learning would have to be modeled on a different spectrum. We focus on the multisensory imagery effect (as described in our discussion), which explains how humans simplify the understanding of absent stimuli when reconstructing it based on an expected idea. For example, while reading a text we can imagine the prosodic characteristics of a child's voice or an adult voice, for example. But would be very hard to recreate the exact tone, pitch, and accent of a specific child or a specific adult. The multisensory imagery effect allows humans to fast-process multisensory information, and, most importantly, to modulate different decision-making processes when one of the sensory information is absent or corrupted. Our approach for expectation learning would not be feasible for precise reconstruction of information, but for prototype learning of multisensory concepts. As demonstrated by our experiment, our solution would fit the modeling of two-way sensory mapping information. The implication is that our model would be able to be used with similar audio/visual tasks, i.e. as prosodic characteristics of voice and face expressions, sounds and objects, and environment sounds and events. Following our findings on the representation-level association, our model would also be suitable for learning of mappings between higher-abstraction level concepts as soon as it can represent such concepts, i.e. an action with a consequence (close the door, expect a loud noise), a sensory feeling with an semantic context (I feel it is warm, so I must reduce the heating level).
>
> 2. R: We focused on demonstrating that the expected learning can be used to improve unisensory recognition for absent stimuli. We decided to use the baselines we chose due to their importance on different audio and visual recognition tasks. Both Inception V3 and SoundNet models still are considered to have established results on different vision [1, 2, 3] and auditory [5, 6, 7] recognition tasks and are used as a basis for comparison. Besides that, our focus was not to produce the benchmarking results for this specific dataset, as this would rely mostly on hyper-parameter exploration search. We focus on the advantages of modeling multisensory expectation, most importantly on the reconstruction of absent stimuli, and proof that it performs very closely (0.6% accuracy on vision and better on audio)  to these deep neural network-based general models.
>
> [1]Kiros, J., Chan, W., & Hinton, G. (2018). Illustrative Language Understanding: Large-Scale Visual Grounding with Image Search. In Proceedings of the 56th Annual Meeting of the Association for Computational Linguistics  (Vol. 1, pp. 922-933).
>
> [2] Jiang, L., Zhou, Z., Leung, T., Li, L. J., & Fei-Fei, L. (2018, July). MentorNet: Learning data-driven curriculum for very deep neural networks on corrupted labels. In International Conference on Machine Learning (pp. 2309-2318).
>
> [3] Zeng, X., Ouyang, W., Yan, J., Li, H., Xiao, T., Wang, K., ... & Zhou, H. (2018). Crafting gbd-net for object detection. IEEE transactions on pattern analysis and machine intelligence, 40(9), 2109-2123.
>
>
> [5] Kumar, A., Khadkevich, M., & Fügen, C. (2018, April). Knowledge transfer from weakly labeled audio using convolutional neural network for sound events and scenes. In 2018 IEEE International Conference on Acoustics, Speech and Signal Processing (ICASSP) (pp. 326-330).
>
> [6] Jansen, A., Plakal, M., Pandya, R., Ellis, D. P., Hershey, S., Liu, J., ... & Saurous, R. A. (2018, April). Unsupervised learning of semantic audio representations. In 2018 IEEE International Conference on Acoustics, Speech and Signal Processing (ICASSP) (pp. 126-130).
>
> [7] Arandjelovic, R., & Zisserman, A. (2017, October). Look, listen and learn. In 2017 IEEE International Conference on Computer Vision (ICCV) (pp. 609-617).
>
> 3. R: The assumption is correct. To learn the bindings needed for the expectation association, we need strongly paired multisensory data. However, once this is learned, the model can be fine-tuned using unisensory stimuli only, as it would reconstruct the absent stimuli using the expectation association. Our experiments are entirely based on the reconstruction of absent stimuli. Regarding the noisy information, the GWR learns prototype neurons by adding (growing) and removing(shrinking) neurons that do not match the perceived data distribution. If noisy data would be present, the network would create specific prototype neurons to model the noise. However, if the noisy information is not abundant on the dataset, these neurons would not be as activated as the non-noisy neurons and would be removed.
>
> 4. Questions to authors and typos R: Thank you for your feedback. We performed a review on the paper to remove typos and improve the general language aspects.

---

> > ### Comment · AnonReviewer3 · 2018-12-06
> > **on results and evaluation**
> >
> > I thank the authors for the effort put into the rebuttal.
> >
> > 1. good explanations.
> >
> > 2. Regarding results, your rebuttal states that "our focus was not to produce the benchmarking results for this specific dataset". However, your abstract states that "the proposed model presents state-of-the-art performance in representing and classifying unisensory stimuli". Can you please clarify? If using additional modalities does not improve performance as compared to using just the one existing modality, then why should researchers go through the trouble to collect additional data from other modalities and building models that could possibly be sensitive to noise in the other modalities?

---

> > > ### Author Response · Authors · 2018-12-11
> > > **Regarding the results Statement**
> > >
> > > Dear Reviewer,
> > >
> > > 1 - Thank you for the positive comments, I hope we could improve the general understanding of our contribution.
> > >
> > > 2 - Our method differs from current state-of-the-art results in three ways: 1) we provide an unsupervised coincident learning mechanism - which implies the model can learn in an online manner from correlated audio/visual data. That is much more appropiate in real-world scenarios, then most of the strongly supervised and audio/visual decoupled solutions (the ones which artificially associated audio and vision to improve recognition). 2) Our experiments provide an absent-stimuli recognition capability - This implies that even in the absence of one stimulus, the model still demonstrates considerable recognition capabilities by reconstructing the absent stimulus. We do not focus on an expensive hyper-parameter tunning to improve our model but rely heavily on the expectation mechanism we presented. The classification accuracy, however, is close to the state of the art. Our evaluation provides a novel objective measure based on classification for the quality of the absent stimuli representation. Finally, 3), our model does not need to be completely re-trained to be able to learn new audio/visual associations, as the GWR can adapt to novel data on an online fashion.
> > >
> > > Regarding applicability, we do not position our model as a substitute to fully supervised pre-trained unisensory recognition models, in particular when structured, strongly labeled and uniformly distributed data is available. However, when taking into consideration natural data with weak labels, such as the ones present on the AudioSet corpus, our model takes advantage from the natural coincidence of audio/visual occurrence and uses it for learning novel information - from any of the two modalities. Experimenting, analyzing and stating that our model is competitive with state-of-the-art solutions underpins that our model is applicable to real-world problems.

---

### Official Review · AnonReviewer2 · 2018-11-06
**unsupervised approach to learning multi sensory binding with expectation learning**

**Rating:** 4
**Confidence:** 4

**Review:**

The paper develops a multi sensory model that binds audio/visual features using a self-organizing binding learning layer that takes as input the latent layers of two separate auto-encoders.  The binding is learned in an unsupervised way through Hebbian learning, which is an interesting way to implement temporal binding.

My concerns are as below:

- How is the accuracy computed for the experimental data? Is it done moment-by-moment (per frame of the video?)? And if so, how is the accuracy of the audio assessed, since it's sampled at a much higher frequency?

- Fig 2 should be a bar graph, since each class of animal is categorical, and the accuracies should be shown with standard deviations.

- Abstract is written in a very confusing way that does not make clear succinctly what are the contributions of the paper. Generally, the paper may have been assembled in some haste.

---

> ### Author Response · Authors · 2018-11-26
> **Improved general readability of the paper**
>
> Thank you for your comments. We updated the manuscript to make our contribution more understandable. Mostly, we focused on the improvement of the abstract, introduction and experimental setup. We changed only the necessary for a better understanding of our paper and did not make any further modification on our results, or results' interpretations. Regarding your concerns, we updated the manuscript as follows:
>
> - How is the accuracy computed for the experimental data? Is it done moment-by-moment (per frame of the video?)? And if so, how is the accuracy of the audio assessed, since it's sampled at a much higher frequency?
>
> R: Each video has one label, so the accuracy is calculated video-wise. To learn the expectation bindings, we associate each frame of the video with 12.5ms of audio (as explained on Session 2.3 - Self-Organizing Temporal Binding). The GWR has a context of 4 neurons, meaning that each neuron represents the association of 4 consecutive frames and 50ms of audio. The recognition of the stimuli is made using only unisensory stimuli, we do not claim to have multisensory recognition. Our experimental result is based on the recognition of absent stimuli: we present 50ms of audio and obtain the encoded representation for 4 frames, and vice-versa, with 4 frames we recreate 50ms of audio.
>
> For vision recognition, we classify all the frames of a video and use a voting scheme to make a decision for one label. For auditory recognition, we follow the same scheme, recognizing each chunk of 50ms of audio and use a voting scheme to make a decision for one label.
>
> We updated this information in our paper.
>
>
> - Fig 2 should be a bar graph, since each class of animal is categorical, and the accuracies should be shown with standard deviations.
>
> R: We updated Fig.2 to display the data as a bar graph.
>
> - The abstract is written in a very confusing way that does not make clear succinctly what are the contributions of the paper. Generally, the paper may have been assembled in some haste.
>
> R: We updated the abstract to make it easier to understand our contribution. As mentioned above, we also updated the paper to make the readability easier.

---

### Meta-Review · Area_Chair1 · 2018-12-14

**Confidence:** 4
**Recommendation:** Reject

**Metareview:**

The submission introduces a model that does learning of multisensory representations (by predicting one from the other), with an autoencoder structure. Generally, the reviewers liked the overall idea of the work, but found the clarity lacking, the evaluation insufficient (and not particularly state of the art), the requirement for paired training data quite limiting and the choices (VAE sometimes, autoencoder other times) somewhat ad hoc.